**Data Availability Statement:** All relevant data are within the paper and its Supporting information files.

**Funding:** This work was supported by NHLBI: grant R01HL113508 (LS). The funders had no role

# Incidence and risk factors of COVID-19 associated pneumothorax

**Muhanad Taha[1], Morvarid Elahi[1,2], Krista Wahby[3], Lobelia Samavati**[ID][1,2]*

**1** Department of Internal Medicine, Division of Pulmonary, Critical Care and Sleep, Wayne State University School of Medicine and Detroit Medical Center, Detroit, Michigan, United States of America, **2** Center for Molecular Medicine and Genetics, Wayne State University School of Medicine, Detroit, Michigan, United States of America, **3** Department of Pharmacy, Harper University Hospital and Detroit Medical Center, Detroit, Michigan, United States of America

* ay6003@wayne.edu

## Abstract

### Background

Pneumothorax has been increasingly observed among patients with coronavirus disease-2019 (COVID-19) pneumonia, specifically in those patients who develop acute respiratory distress syndrome (ARDS). In this study, we sought to determine the incidence and potential risk factors of pneumothorax in critically ill adults with COVID-19.

### Method

This retrospective cohort study included adult patients with laboratory-confirmed SARS-CoV-2 infection admitted to one of the adult intensive care units of a tertiary, academic teaching hospital from May 2020 through May 2021.

### Results

Among 334 COVID-19 cases requiring ICU admission, the incidence of pneumothorax was 10% (33 patients). Patients who experienced pneumothorax more frequently required vasopressor support (28/33 [84%] vs. 191/301 [63%] P = 0.04), were more likely to be proned (25/33 [75%] vs. 111/301 [36%], P<0.001), and the presence of pneumothorax was associated with prolonged duration of mechanical ventilation; 21 (1–97) versus 7 (1–79) days, p<0.001 as well as prolonged hospital length of stay (29 [9–133] vs. 15 [1–90] days, P<0.001), but mortality was not significantly different between groups. Importantly, when we performed a Cox proportional hazard ratio (HR) model of multivariate parameters, we found that administration of tocilizumab significantly increased the risk of developing pneumothorax (HR = 10.7; CI [3.6–32], P<0.001).

### Conclusion

Among 334 critically ill patients with COVID-19, the incidence of pneumothorax was 10%. Presence of pneumothorax was associated with prolonged duration of mechanical ventilation and length of hospital stay. Strikingly, receipt of tocilizumab was associated with an increased risk of developing pneumothorax.

in study design, data collection and analysis, decision to publish, or preparation of the manuscript.

**Competing interests:** The authors have declared that no competing interest exist.

## Introduction

The SARS-CoV-2 infection has caused the COVID-19 pandemic, a highly variable clinical syndrome ranging from asymptomatic carriers to acute respiratory failure and high mortality [1, 2]. Epidemiological data indicates that 6 to 10% of COVID-19 patients will develop severe respiratory symptoms and will require intensive care unit (ICU) admission, which is associated with poor outcome [3]. COVID-19 associated lung injury and respiratory failure often results in prolonged need for non-invasive or invasive oxygen therapy and is linked to an extended length of ICU stay. The underlying pathogeneses of COVID-19 induced respiratory failure include damage to the angiotensin-converting enzyme-2 (ACE2) receptor of endothelial and epithelial cells, resulting in cellular injury. Additionally, damage to lung epithelial and endothelial ACE2 receptors may result in impaired cellular repair mechanisms [4, 5]. COVID-19 pneumonia, unlike other viral pneumonias, seems to be disproportionally associated with elevated incidence of thromboembolism and pneumothorax [6, 7].

Pneumothorax impairs ventilation and oxygenation and can manifest spontaneously or due to barotrauma as a complication of mechanical ventilation that can be associated with subcutaneous emphysema and pneumomediastinum or pneumopericardium. Pneumothorax can complicate mechanical ventilation especially in patients with COVID-19 associated pneumonia, leading to a prolonged ICU stay. In a recent meta-analysis of 1,814 invasively ventilated COVID-19 patients, barotrauma occurred in one out of six patients (14.7%) and it was associated with increased mortality [8]. The incidence of pneumothorax in mechanically ventilated COVID-19 patients appear to be higher than in conventional acute respiratory distress syndrome (ARDS) or other viral pneumonias [7–9]. It appears that over time the frequency of reported incidence of pneumothorax is increasing during COVID-19 pandemic [10]. Although it is possible that new variants are contributing to higher rates of pneumothorax, the role of pharmacotherapies targeting the body's innate defense mechanisms must be considered. Many repurposed drugs explored to treat COVID-19 modulate inflammation, some of which significantly alter the immune response, including viral clearance, host cellular responses, or cellular repair mechanisms. Besides the corticosteroids, tocilizumab (TCZ) is one of the most utilized drugs to modulate the hyper-inflammatory responses in COVID-19 infection [11, 12]. Tocilizumab is a monoclonal antibody against interleukin-6 receptors, approved to treat rheumatoid arthritis with a relatively good safety profile [13]. Despite negative results of TCZ efficacy in earlier studies in treatment of severe COVID-19 [14], the Food and Drug Administration (FDA) has recently approved the use of TCZ based on more recent clinical studies [15, 16].

As clinicians continue to treat critically ill COVID-19 patients in the ICU, it is important to understand the risk factors associated with development of pneumothorax. In this study, we sought to determine the incidence, risk factors and the clinical consequences of pneumothorax in COVID-19 patients requiring ICU admission.

## Materials and methods

In this retrospective, cohort, observational single-center study, we examined the incidence of pneumothorax in patients with COVID-19 positive, who were admitted to the intensive care unit (ICU) for respiratory distress at Detroit Receiving Hospital and Harper University Hospital in Detroit Medical Center in Detroit, Michigan. After receiving approval from the Institutional Review Board at Wayne State University, patients aged $\geq$ 18 years old requiring ICU admission from May 2020 to February 2021, with laboratory-confirmed SARS-CoV-2 infection, diagnosed by reverse-transcriptase–polymerase-chain-reaction (RT-PCR) of nasopharyngeal specimen, were included. Demographic and clinical characteristics were obtained

from electronic medical records. The diagnosis of barotrauma was established using portable chest radiograph. Patients were divided into two cohort, those with and those without pneumothorax, and were followed from admission to discharge or death. Patients with iatrogenic pneumothorax, or those requiring thoracentesis or chest tube drainage for pleural effusion; or receiving extracorporeal membrane oxygenation (ECMO) were excluded. Respiratory support, including supplemental oxygen therapy and mechanical ventilation, were applied according to the best practice of using standard lung protective ventilation strategies adopted in ARDS network trial [17]. COVID-19 specific therapies were provided per our institution guidance protocol (S1 File) including anticoagulation therapy. According to our institution protocol, all patients with COVID-19 should receive full anticoagulation therapy if D-dimer >3. Tocilizumab required approval from our Infectious Diseases service and a weight-based dose was used, consistent with the Infectious Diseases Society of America Guidelines [18]. Charlson Comorbidity Index Score and Acute Physiology and Chronic Health Evaluation (APACHE) II were calculated as previously described [19, 20].

Study is approved by human investigation committee at the Wayne State University. The committee waived the requirement for informed consent. All methods were performed in accordance with the human investigation ethical guidelines and regulations by the local IRB (protocol No = IRB-20-04-2037) at Wayne State University.

## Statistical analysis

Parametric continuous variables were represented as mean with standard deviation. Nonparametric continuous samples were represented as median with interquartile range (IQR). Categorical data were presented as frequencies and percentages with inference by Pearson's chi-square test. Statistical significance was defined by a p-value <0.05. For ventilator settings, averages of all values recorded throughout the ICU-admission were calculated for each patient. Then, the average for the entire ICU stay was calculated and used for final statistical analyses. Univariate analyses of all independent variables were performed to compare the two cohorts by using independent-samples t-test, Mann-Whitney U test for nonparametric, or Pearson's chi-square test, as appropriate. All independent variables with p-value of less than 0.05 in the univariate analysis were enrolled into multivariate analyses. Cox proportional hazards regression model with Enter method was then performed to evaluate several co-factors simultaneously and identify the predictors of pneumothorax. Kaplan-Meier Survival curves were plotted to compare survival between two groups based on Tocilizumab therapy. Log rank test (Mantel-Cox) was used to compare the survival distributions of the two cohorts at 25 days. SPSS statistical software (Version 27, Chicago, IL) was used.

## Results

Out of 349 patients admitted to the ICU with laboratory-confirmed COVID-19, 334 patients with a mean age of 61 ± 14 years, 188 (56%) males, 245(73%) African-Americans were included in the analyses. Five subjects were excluded as they developed pneumothorax as a complication of invasive procedures (3 central lines, 2 chest tube for empyema). Eleven subjects, who received ECMO were also excluded. Overall, the incidence of pneumothorax was 10% (33/334) followed by pneumomediastinum 20/334 (6%) and subcutaneous emphysema 9/334 (3%). The mean duration of ICU stay prior to pneumothorax was 15.7 days, and the mean duration of mechanical ventilation prior to barotrauma was 10.3 days. There were 27/33 (81%) patients who required chest tube insertion to treat pneumothorax. Table 1 summarizes patients' clinical characteristics and the comparison of clinical features between the two cohorts. Most patients were admitted first to an acute care unit where medical treatment was

**Table 1. Subjects characteristics.**

| Characteristics | All Patients n = 334 | Non- pneumothorax n = 301 (90%) | Pneumothorax n = 33 (10%) | p-value |
|---|---|---|---|---|
| **Age, Years (mean ± SD)** | 61 ± 14 | 61 ± 14 | 59 ± 13 | 0.61 |
| **Sex, n (%)** | | | | 0.47 |
| Female | 148 (44) | 132 (44) | 16 (49) | |
| Male | 188 (56) | 169 (56) | 17 (51) | |
| **Known ethnicity, n (%)** | | | | 0.68 |
| African—American | 245 (73) | 225 (74) | 20 (60) | |
| White/Caucasian | 88 (26) | 75 (25) | 13(40) | |
| Hispanic | 1 (0.2) | 1 (0.3) | 0 | |
| **BMI, kg · m$^{-2}$, median(IQR)** | 31 (15–85) | 30 (15–85) | 32 (22–62) | 0.49 |
| **Smoker, n (%)** | 80 (24) | 73 (24) | 7 (21) | 0.43 |
| **Comorbidities, n (%)** | | | | |
| COPD [a] | 75 (22) | 68 (22) | 7 (21) | 0.92 |
| Asthma | 34 (10) | 31 (10) | 3 (9) | 0.55 |
| Hypertension | 266 (79) | 239 (79) | 27 (81) | 0.48 |
| Diabetes mellitus | 200 (60) | 177 (58) | 23 (69) | 0.15 |
| Coronary artery disease | 67 (20) | 62 (20) | 5 (15) | 0.31 |
| Chronic heart failure | 68 (20) | 65(21) | 3 (9) | 0.16 |
| Cerebrovascular accident | 49 (14) | 46 (15) | 3 (9) | 0.25 |
| Chronic kidney disease | 56 (16) | 54 (18) | 2 (6) | 0.06 |
| ESRD [b] | 43 (12) | 36 (12) | 7 (21) | 0.11 |
| Malignancy | 45 (13) | 42 (14) | 3 (9) | 0.31 |
| Autoimmune disease | 18 (5) | 17 (5) | 1 (3) | 0.44 |
| HIV [c] infection | 4 (0.5) | 2 (0.6) | 2 (6) | **0.05** |
| **Charlson Comorbidity Index, mean ± SD** | 3 ± 2 | 3 ± 2 | 2 ± 2 | 0.06 |
| **APACHE II [d] Score, mean ± SD** | 10 ± 5 | 10 ± 5 | 10 ± 4 | 0.19 |

[a] Chronic obstructive pulmonary disease,

[b] End stage renal disease,

[c] Human immunodeficiency virus,

[d] Acute Physiology and Chronic Health Evaluation II.

initiated. No significant differences were seen in demographics, comorbidities, or APACHE II scores on admission between the pneumothorax and non-pneumothorax cohorts, except for human immunodeficiency virus infection (HIV), which was significantly higher in the pneumothorax cohort (p = 0.05) (Table 1). The univariate analysis performed between the pneumothorax and non-pneumothorax subgroups is shown in Table 2. Median with IQR of length of hospital stay for all 334 patients was 16 (1–133) days and median length of ICU stay was 11 days (IQR = 1–100). The pneumothorax group had a significantly longer hospital and ICU stay. During their ICU stay, 299 (89%) individuals required invasive ventilation, 28 (8%) needed non-invasive ventilation, and 7 subjects (2%) required high flow nasal cannula (HFNC). Median duration of mechanical ventilation was 8 (1–97) days. Among 334 patients, 136 (40%) subjects were placed in the prone positioning during ICU stay. Subjects in the pneumothorax cohort were more likely to require prone positioning either before or after pneumothorax (75% vs 36%; p<0.001) (Table 1). Most patients developed pneumothorax while on invasive mechanical ventilation, except two patients, who developed pneumothorax while receiving HFNC or non-invasive mechanical ventilation. There were no significant differences

**Table 2. Respiratory supports, medical managements and outcome.**

| Features | All Patients n = 334 | Non-pneumothorax n = 301 | Pneumothorax n = 33 | p-value |
|---|---|---|---|---|
| **Total length of hospital stay\*, median(IQR)** | 16 (1–133) | 15 (1–90) | 29 (9–133) | **<0.001** |
| **ICU length of stay\*, median(IQR)** | 11 (1–100) | 10 (1–79) | 23 (3–100) | **<0.001** |
| **Maximal respiratory support, n (%)** | | | | |
| High flow nasal cannula | 7 (1) | 6 (2) | 1 | 0.08 |
| Non-invasive mechanical ventilation | 28 (8) | 26(7) | 2(1) | **0.04** |
| Invasive mechanical ventilation | 299 (89) | 269 (78) | 30 (94) | **0.01** |
| **Duration of Invasive ventilation\*, median(IQR)** | 8 (1–97) | 7 (1–79) | 21 (1–97) | **<0.001** |
| **Proning, n (%)** | 136 (40) | 111 (36) | 25 (75) | **<0.001** |
| **Mechanical ventilation parameters, mean ± SD** | | | | |
| Initial respiratory rate | 22 ± 7 | 22 ± 7 | 22 ± 7 | 0.83 |
| Average respiratory rate | 22 ± 3 | 22 ± 3 | 23 ± 4 | **0.04** |
| $FiO_2$ (%) | 71 ± 20 | 71 ± 20 | 76 ± 16 | 0.32 |
| Tidal volume mL/kg PBW$ | 5.9±2.1 | 6±1.8 | 6.2±1.9 | 0.6 |
| Positive end-expiratory pressure (cm $H_2O$) | 9.5 ± 3.4 | 9.5 ± 3.4 | 9.9 ± 3.2 | 0.52 |
| Peak inspiratory pressure (cm $H_2O$) | 28 ± 6 | 28 ± 6 | 30 ± 5 | 0.07 |
| Plateau pressure (cm $H_2O$) | 24 ± 5 | 24 ± 5 | 25 ± 4 | 0.23 |
| **Laboratory values, median(IQR)** | | | | |
| Initial C-reactive protein (mg/L) | 132 (5–1800) | 132 (5–1800) | 139 (25–620) | 0.28 |
| Initial ferritin (ng/mL) | 518 (11–7500) | 517 (11–7500) | 575 (20–7500) | 0.64 |
| **Treatment, n (%)** | | | | |
| Glucocorticoids | 251 (75) | 220 (73) | 31 (94) | **0.004** |
| Hydroxychloroquine | 188 (56) | 179 (59) | 9 (27) | **<0.001** |
| Tocilizumab | 88 (26) | 66(22) | 22 (66) | **<0.001** |
| Remdesivir | 33 (9) | 25 (8) | 8 (24) | **0.01** |
| Vasopressors | 219 (65) | 191 (63) | 28 (84) | **0.04** |
| Therapeutic anticoagulant | 202 (60) | 176 (58) | 26 (78) | **0.01** |
| **Ventilator dependent, n (%)** | 52 (15) | 42 (14) | 10 (30) | 0.25 |
| **Mortality, n (%)** | 139 (41) | 129 (42) | 10 (30) | 0.26 |

\*Days,

$ predicted body weight

in ventilator parameters between two groups, except the average RR. Similarly, there were no significant differences in the initial C-reactive protein (CRP) or ferritin levels between two groups. However, there were significant differences in the COVID-19 medical management between two groups: overall 75% (251/334) of patients received glucocorticoids; 73% of non-pneumothorax group, compared to 94% of pneumothorax group, (p = 0.004). Overall, 188/334 (56%) subjects received hydroxychloroquine, 59% in the non- pneumothorax group vs. 27% in the pneumothorax group, (p<0.001). Eighty-eight (26%) patients received TCZ, 22% in the non- pneumothorax group vs. 66% in the pneumothorax group. Similarly, 8% of the non-pneumothorax compared to 24% of patients in the pneumothorax group received remdesivir. Vasopressor requirements between two groups were also significantly different (p = 0.04). Additionally, based on our institution guidance (S1 File), the pneumothorax group was more likely to receive prophylactic full anticoagulation therapy, 78% versus 58%, p = 0.01. There were no significant differences in terms of mortality or dependency on ventilator as reflected by the requirement of tracheostomy placements and long-term care facility transfer (Table 2).

**Table 3. Hazard risk of pneumothorax.**

| Variable | Hazard ratio (95% CI) | p-value |
|---|---|---|
| ICU length of stay | 0.92 (0.88–0.97) | **0.005** |
| Duration of invasive ventilation | 1.003 (0.95–1.05) | 0.90 |
| Average Respiratory rate | 1.01 (0.90–1.14) | 0.76 |
| Proning: yes vs. no | 1.02 (0.38–2.76) | 0.96 |
| Glucocorticoids: yes vs. no | 0.58 (0.10–3.35) | 0.54 |
| Hydroxychloroquine: yes vs. no | 1.12 (0.40–3.14) | 0.81 |
| Tocilizumab: yes vs. no | 10.7 (3.60–32.0) | **<0.001** |
| Remdesivir: yes vs. no | 1.78 (0.68–4.66) | 0.23 |
| Vasopressors: yes vs. no | 1.64 (0.43–6.18) | 0.46 |
| Therapeutic anticoagulant: yes vs. no | 0.69 (0.25–1.92) | 0.48 |

## Hazard risks for pneumothorax

To evaluate for potential risk factors for pneumothorax, we performed a Cox proportional hazard regression (HR) model and evaluated several co-factors simultaneously to identify the best model to predict the presence of pneumothorax. To build the regression model, we selected cofactors, which were significantly different (p< 0.05) in univariate analysis between the two cohorts. The hazard probability of predictors lies in the range 0 to 1. We adjusted HR based on age, gender and BMI (Table 3) As shown, the ICU length of stay [HR = 0.92; CI (0.88–0.97), p = 0.005]; and TCZ therapy [HR = 10.7; CI (3.60–32.0), p<0.001] were significantly associated with pneumothorax. Although corticosteroids showed a HR of 0.58, the p value was not significant. Fig 1A shows the cumulative hazard ratio between cohorts who received TCZ

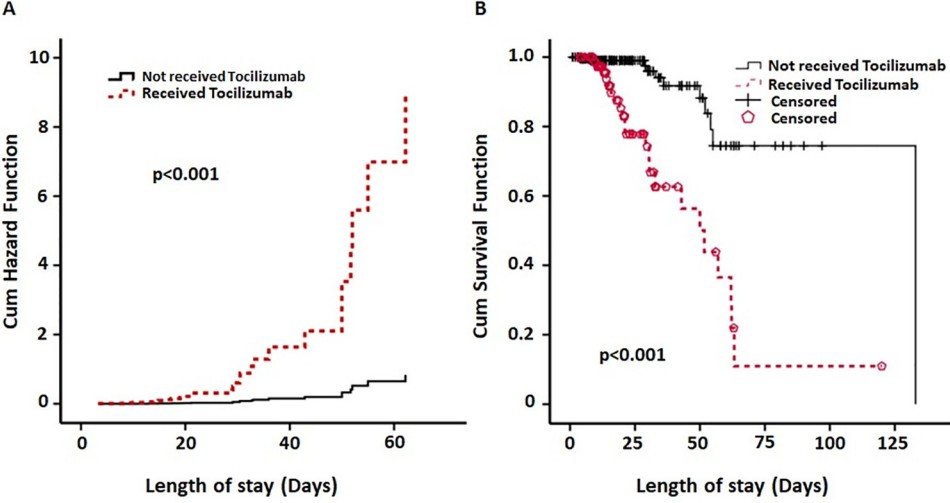

**Fig 1.  A**. Cox regression curve shows cumulative hazards function of Tocilizumab in developing pneumothorax during time of admission after adjusting for significant covariates. The black curve represents patients who did not receive tocilizumab and the red dotted curve represents patients who received tocilizumab. There was a significant difference (chi square test, p<0.001) between subjects who received tocilizumab and who didn't. **B**. Kaplan-Meier survival curve shows cumulative probability of survival function of pneumothorax in two cohorts of tocilizumab and non-tocilizumab with known outcome up to 25 days of hospitalization. The black curve represents patients who did not receive tocilizumab and red dotted curve depicts patients who received tocilizumab. Survival function curves of pneumothorax are statistically different between two group based on receiving tocilizumab (log rank test, p<0.001) during time of admission.

therapy and who did not. The HR curve of pneumothorax in the TCZ group was increased around day 30 of hospital stay (Fig 1A). Fig 1B shows the Kaplan–Meier Curves depicting the survival function of pneumothorax in two cohorts based on receipt of TCZ therapy.

## Discussion

In this study, we found an overall incidence of pneumothorax of 10% among critically ill COVID-19 patients requiring ICU admission and invasive and/or non-invasive mechanical ventilation despite following the lung protective strategy. Development of pneumothorax in COVID-19 patients was associated with poor clinical outcomes and increased length of ICU stay. Although the This data corroborates recent publications evaluating the incidence and outcomes of pneumothorax in COVID-19 patients [7, 21, 22].

For the past two decades, lung-protective ventilatory strategies have become the standard of the care in ARDS management [23]. These strategies include low TV (4–6 mL/kg predicted body weight [PBW]), to achieve the lowest possible PEEP, peak and plateau pressures (<30 cm $H_2O$) and avoid barotrauma [17]. Our study showed that tidal volume, PEEP, and peak and plateau pressures were not significantly different between patients who developed pneumothorax and those who did not. Furthermore, it has been described that the increased work of breathing in COVID-19 patients may induce Self- Inflicted Lung Injury (PSILI) [24], which may be a potential risk factor for pneumothorax. Although, our study showed a significant difference between groups in terms of mean RR, most patients who developed pneumothorax were paralyzed at the time of pneumothorax development, therefore, the PSILI less likely to explain the development of pneumothorax.

SARS-CoV-2 utilizes ACE2 receptors to gain entry into host cells, and data suggests that-ACE2 serves a protective function in the alveolar epithelial repair process [5]. The damage of ACE2 by the SARS-CoV-2 virus may contribute to increased susceptibility to lung injury, including pneumothorax. However, the reported incidence of COVID-19associated pneumothorax is steadily increasing. Thus, it is warranted to seek other potential mechanisms of pneumothorax beyond the ventilatory settings or factors related to the pathogen itself, including the contribution of pharmacological therapies on cellular injury and repair. Unlike treatment of any viral infection or classical ARDS, treatment of COVID-19 associated respiratory failure led to initiation of numerous unprecedented novel treatments. As the ongoing pandemic continues, it is important to evaluate the potential interaction of these novel treatments on outcome and complications. Thus, we investigated the risk factors of pneumothorax using multivariate analysis and cox proportional regression hazard model and found that TCZ significantly associated with a higher risk of pneumothorax in COVID-19 patients.

Severe COVID-19 disease is linked to a hyper-inflammatory state based on high fever, elevated CRP, ferritin and various serum cytokines, including IL-6, tumor necrosis factor and IL-1β and others [25, 26]. Several studies suggest that IL-6 is a key cytokine associated with severity of COVID-19 disease [25, 27]. IL-6 is produced by almost all immune cells in response to infection and tissue damage and has various pleiotropic actions including B-cell activation, neutrophil and macrophage recruitments, and increased vascular permeability [28]. Therefore, inhibition of IL-6 was proposed as a potential therapeutic target when treating patients with COVID-19. Tocilizumab (TCZ) is a humanized antibody that binds to the IL-6 receptor, inhibiting its action [29]. Based on the efficacy of TCZ in cytokine release syndrome induced by CAR T-cell therapy, and potential role of IL-6 in the pathophysiology of COVID-19 ARDS, TCZ was promoted as a repurposed drug. Initial observational studies suggested that TCZ reduced mortality in COVID-19 patients [30–32]. However, randomized trials studying the efficacy of TCZ in hospitalized patients with COVID-19 failed to show a mortality benefit [33,

34]. The study by Veiga et al., was stopped early due to excessive mortality in the TCZ group at day 15 [34]. A meta-analysis of these early studies also failed to show a 28-day mortality benefit when using TCZ with a pooled RR of death of 1.09 (95% CI, 0.80–1.49). These studies primarily included patients, not requiring invasive or non-invasive mechanical ventilation, and therefore, recruited patients had much lower risk for pneumothorax than our study subjects. In addition, these studies were performed before the RECOVERY dexamethasone study [35] was published, so that the standard of care as it relates to administration of dexamethasone was different than current practice and rates of glucocorticoid administration were generally less than 25%. Safety events were assessed in some of these studies, however, pneumothorax were either not reported [15, 31], or showed no difference between TCZ patients and control [33, 34].

COVACTA was the first randomized controlled trial released publicly and included 452 patients (294 TCZ/144 control) hospitalized with severe COVID-19 pneumonia and treated with TCZ 8 mg/kg or placebo [11]. No significant difference was found in 28-day outcomes using a 7-category ordinal scale or mortality [11]. When looking specifically at the subset of patients receiving mechanical ventilation in this study, no benefit was shown by giving TCZ [23]. There was a high rate of adverse events in both the control and TCZ groups (81.1% vs. 77.3%, respectively.) but there was no mention of pneumothorax. Most studies have focused on anaphylaxis and infection as the main safety outcomes related to TCZ. It is important to note that there is a significant heterogeneity among TCZ studies, in terms of severity of COVID-19 disease, ICU requirements, or concomitant treatments. Studies have shown mixed results in critically ill COVID-patients requiring intensive care [12, 31]. After release of the RECOVERY trial using low-dose dexamethasone, the standard of care for treating hypoxic patients with COVID-19 changed and subsequent studies assessing TCZ combined with corticosteroids began to show improved outcomes [11, 16, 36, 37]. The REMAP-CAP trial had the highest percentage of patients on non-invasive and invasive ventilation and high rates of corticosteroid use (>80%), which is similar to our study population. They showed a significant reduction in 90-day mortality and more organ support-free days, but no significant adverse events [16]. To date, no significant differences in adverse events have been reported in studies comparing TCZ to standard care, but the care in most of these studies was different, in term of administration of dexamethasone, early anti-viral treatment with remdesivir, early monoclonal antibodies and prophylactic, not therapeutic anticoagulation in critically ill patients.

Our study shows that pneumothorax occurs relatively late (median 30 and mean 16 days from admission) in the course of disease, when CRP and fever were already down trending. Therefore, we postulated that dysregulation of lung repair may be important in pneumothorax development. This raises the question whether inhibition of IL-6 results in deleterious effects related to epithelial healing and cellular repair. IL-6 plays a key role in viral clearance, as well as angiogenesis, and therefore, inhibition of IL-6 could potentially cause harm through reduced viral clearance and interruption of the innate repair processes. In patients with severe lung injury, the remodeling and repair process may be an important piece in preventing pneumothorax. In a study examining biomarkers in COVID-19, survivors of COVID-19 had higher levels of epithelial growth factor, suggesting that alveolar regeneration and repair may be key features of lung recovery. Mechanisms to repair the alveolar epithelial barrier and restore a competent monolayer are initiated immediately in response to tissue injury. Initial epithelial repair events in ALI include proliferation, spreading, and migration of ATII cells to cover the denuded alveolar basement membrane [38]. Some insights in the role of IL-6 are provided by an animal model of influenza virus using IL-6 deficient mice showing that IL-6 ameliorates ALI after infection [39]. Similar effects of IL-6 have been shown by other animal studies [40]. Tocilizumab is a potent inhibitor of IL-8 and IL-6 and inhibits angiogenesis, however to what extent it inhibits EGF in COVID-19 is not known. Additionally, it is unclear what role, if any,

the combination therapies like tocilizumab and dexamethasone have on the lung repair and recovery from respiratory failure, and whether antiviral treatment with remdesivir is important to facilitate viral clearance in patients whose immune response is impaired. Pneumothorax may be one identifiable example of how these drugs may interfere with the lung repair mechanisms. To our knowledge, this is the first study to raise concern about this serious adverse event of tocilizumab therapy in COVID-19.

Our study has certain noteworthy limitations. First, our study, like any other observational study, is prone to bias due to unmeasured confounders. Second, data was obtained from critically ill patients with relatively higher comorbid conditions that may raise the possibility of selection bias. Therefore, larger studies in critically ill patients are needed to assess the utility of TCZ. Third, the fact that receiving TCZ was guided by specific indications according to our institutional guidelines may create an indication bias. Fourth, missing and misclassification of data are possible because data were manually extracted. However, cox regression analysis did not include any variable with more than 10% missing data.

## Conclusions

Pneumothorax is frequent in COVID-19 and is associated with poor outcomes. In our cohort, tocilizumab therapy was associated with increased HR for pneumothorax and increased length of stay in the ICU.

## Supporting information

**S1 File. Institutional guidelines for Tocilizumab and anticoagulation therapy.** List of inclusion and exclusion criteria for tocilizumab therapy in our institution.
(DOCX)

## Author Contributions

**Conceptualization:** Lobelia Samavati.

**Data curation:** Muhanad Taha, Morvarid Elahi, Krista Wahby.

**Formal analysis:** Lobelia Samavati.

**Funding acquisition:** Lobelia Samavati.

**Investigation:** Muhanad Taha, Morvarid Elahi, Krista Wahby.

**Supervision:** Lobelia Samavati.

**Writing – original draft:** Muhanad Taha, Morvarid Elahi, Krista Wahby.

**Writing – review & editing:** Krista Wahby, Lobelia Samavati.

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
