## [Decision Letter · Decision Letter 0]

3 Jun 2022

PONE-D-22-10994Incidence and Risk Factors of COVID-19 Associated PneumothoraxPLOS ONE

Dear Dr. Samavati,

Thank you for submitting your manuscript to PLOS ONE. After careful consideration, we feel that it has merit but does not fully meet PLOS ONE’s publication criteria as it currently stands. Therefore, we invite you to submit a revised version of the manuscript that addresses the points raised during the review process.

Two reviewers experts in the field underlined the importance of the data also points to be addressed. Specifically, please address the comments related to sample size justification and the statistical analysis

We look forward to receiving your revised manuscript.

Kind regards,

Andrea Cortegiani, M.D.

Academic Editor

PLOS ONE

Journal Requirements:

"This work was supported by NHLBI: grant R01HL113508 (LS)."

5. Thank you for stating the following in the Funding Section of your manuscript: 

"Funding: This work was supported by NHLBI: grant R01HL113508 (LS) and R21 HL148089 (LS)."

We note that you have provided funding information that is not currently declared in your Funding Statement. However, funding information should not appear in the Funding section or other areas of your manuscript. We will only publish funding information present in the Funding Statement section of the online submission form. 

"This work was supported by NHLBI: grant R01HL113508 (LS)."

Reviewers' comments:

Reviewer's Responses to Questions

**Comments to the Author**

1. Is the manuscript technically sound, and do the data support the conclusions?

Reviewer #1: Yes

Reviewer #2: Yes

2. Has the statistical analysis been performed appropriately and rigorously? 

Reviewer #1: Yes

Reviewer #2: Yes

3. Have the authors made all data underlying the findings in their manuscript fully available?

Reviewer #1: Yes

Reviewer #2: Yes

4. Is the manuscript presented in an intelligible fashion and written in standard English?

Reviewer #1: Yes

Reviewer #2: Yes

5. Review Comments to the Author

Reviewer #1: Dear Editor,

Dear Authors,

I read the study entitled: "Incidence and Risk Factors of COVID-19 Associated Pneumothorax with great attention". The study is an observational retrospective investigation finding the "frequency" of pneumothorax in a population largely formed by African – American people.

Finally, the authors found a 10% pneumothorax frequency and, as a secondary result, tocilizumab was associated with it.

1) INTRODUCTION:

In the introduction, the authors comment that in COVID-19, thromboembolism and pneumothorax seem to show an elevated incidence.

In my opinion, you may better talk about "frequence". Second, when you argue about pneumothorax, you should speak about barotrauma. Barotrauma can be further composed of pneumothorax and pneumomediastinum.

It seems so uncommon to not have any pneumomediastinum in your population.

Up to date, what is known in COVID-19 is that pneumothorax is less frequent than pneumomediastinum compared to traditional ARDS. However, the effect of this phenomenon is largely unknown?

See this reference: McGuinness G, Zhan C, Rosenberg N, Azour L, Wickstrom M, Mason DM, Thomas KM, Moore WH, Increased Incidence of Barotrauma in Patients with COVID-19 on Invasive Mechanical Ventilation, Radiology, vol. 297, no. 2, pp. E252-E62, Nov 2020.doi:10.1148/radiol.2020202352.

The currently published scientific literature shows that pneumothorax occurs in 1 to 3% of hospitalized COVID-19 cases, with up to 6% in patients undergoing non-invasive ventilation (NIV) and mechanical ventilation (MV). Therefore, the frequency of pneumomediastinum and pneumothorax during COVID-19 is not well defined, as the available data are limited to case collections and single reports.

In McGuinness's analysis (see ref. above), which compared complications from barotrauma in patients with acute respiratory distress syndrome (ARDS) in VAM, the frequency of pneumothorax is 9%, of pneumomediastinum 10% in COVID- 19. In comparison, in non-COVID-19 patients, it is 12% for PNX and 3% for PMS.

2) INTRODUCTION:

This sentence needs a reference if true:

"It appears that the incidence of pneumothorax is increasing over time with subsequent waves of COVID-19 infection".

Along the line, the authors use pneumothorax and barotrauma indifferently, which should be corrected.

3) METHODS:

This section should be improved and explain inclusion and exclusion criteria and why the grade of pulmonary damage was not reported as lung ultrasound or CT scan scores.

4) STATISTIC

In the statistical section, the power analysis of the first aim is missing.

5) RESULTS

"Subjects in the pneumothorax cohort were more likely to require prone positioning (75% vs 36%; p<0.001)" maybe this is the consequence, not the cause?

"(Table 1). Most patients developed pneumothorax while on invasive mechanical ventilation" however, in the two courts, there were no significant differences in ventilator parameters between the two groups, except the average RR. How do you make the diagnosis of barotrauma? Could barotrauma be present before intubation?

6) RESULTS:

"Additionally, the pneumothorax group was

more likely to receive anticoagulation therapy, 78% versus 58%, p=0.01" how do you modify therapy?

There were no significant differences in mortality. That is not easy to understand.

"Second, we investigated the risk factors of barotrauma using multivariate analysis and the Cox proportional regression hazard model and found that TCZ significantly increased the risk of pneumothorax in COVID-19 patients.

That is a secondary endpoint without power analysis, and an association does not mean that this is true.

In this court, pneumothorax occurs relatively late (median 30 days) in the course

of the disease, when CRP and fever were already down-trending sound like, in the end, the lung has lost compliance and became more susceptible to barotrauma. We have observed in COVID-19 that barotrauma in COVID-19 is an early phenomenon linked to P-SILI without a clear explanation at the moment.

Reviewer #2: Dear authors,

I enjoyed reviewing this interesting paper. The subject is interesting and the paper in well written. I have detailed some comments below.

MAJOR COMMENTS:

1. In the introduction, the authors state that "Pneumothorax occurs when alveoli become damaged and/or ruptured due to elevated peak and plateau pressures, and can manifest as spontaneous pneumothorax, subcutaneous emphysema and pneumomediastinum.". Nonetheless, this is not always true ( doi: 10.1007/s00134-004-2187-7). Moreover, during the COVID-19 pandemic there seemed to be a remarkable increase in pneumomediastinum/subcutaneous emphysema occurrence despite the use of the same protective mechanical ventilation protocol used in non- COVID ARDS patients (https://doi.org/10.1183%2F23120541.00385-2020).This sentence need therefore to be rephrased.

2. Due to what is detailed in the above comment, the term "barotrauma" should be changed throughout the manuscript and abstract, using it only when elevated airway pressure is the cause of the presence of air outside the tracheobronchial tree. In its absence, such a condition should not be referred to as barotrauma, but simply described for what it is (pneumomediastinum, subcutaneous emphysema, pneumothorax)(https://doi.org/10.1183%2F23120541.00385-2020). In your study, tidal volume, PEEP, and peak and plateau pressures were not significantly different between patients who developed pneumothorax and those who did not.

6. PLOS authors have the option to publish the peer review history of their article (what does this mean?). If published, this will include your full peer review and any attached files.

Reviewer #1: **Yes: **Luigi Vetrugno

Reviewer #2: No

---

## [Author Response · Author response to Decision Letter 0]

7 Jul 2022

Point by point response:

We sincerely appreciate all valuable comments and suggestions, which helped us to improve the quality of the article. Our responses to the Reviewers’ comment are described below in a point-to-point manner. Appropriated changes, suggested by the Reviewers, has been introduced to the manuscript (highlighted within the document).

We hope that our manuscript will be acceptable for publication in PLOS ONE journal. 

Journal Requirements:

Reply: We adjusted our manuscript to meet all PLOS ONE's style requirements

Reply: We added an ethics statement in the method section as required above. The statement specify that “the study is approved by human investigation committee at the Wayne State University. The committee waived the requirement for informed consent”:

"This work was supported by NHLBI: grant R01HL113508 (LS)."

Reply: The funders had no role. We included the suggested statement above in our cover letter. 

5. Thank you for stating the following in the Funding Section of your manuscript: 

"Funding: This work was supported by NHLBI: grant R01HL113508 (LS) and R21 HL148089 (LS)."

We note that you have provided funding information that is not currently declared in your Funding Statement. However, funding information should not appear in the Funding section or other areas of your manuscript. We will only publish funding information present in the Funding Statement section of the online submission form. 

"This work was supported by NHLBI: grant R01HL113508 (LS)."

Reply: Information was added in the cover letter.

Reply: We added the ethics statement only in the method section as required above.

We added the ethics statement only in the method section as required above.

Reviewers' comments:

Reviewer's Responses to Questions

Comments to the Author

1. Is the manuscript technically sound, and do the data support the conclusions?

Reviewer #1: Yes

Reviewer #2: Yes

2. Has the statistical analysis been performed appropriately and rigorously?

Reviewer #1: Yes

Reviewer #2: Yes

3. Have the authors made all data underlying the findings in their manuscript fully available?

Reviewer #1: Yes

Reviewer #2: Yes

4. Is the manuscript presented in an intelligible fashion and written in standard English?

Reviewer #1: Yes

Reviewer #2: Yes

5. Review Comments to the Author

Reviewer #1: Dear Editor,

Dear Authors,

I read the study entitled: "Incidence and Risk Factors of COVID-19 Associated Pneumothorax with great attention". The study is an observational retrospective investigation finding the "frequency" of pneumothorax in a population largely formed by African – American people.

Finally, the authors found a 10% pneumothorax frequency and, as a secondary result, tocilizumab was associated with it.

1) INTRODUCTION:

In the introduction, the authors comment that in COVID-19, thromboembolism and pneumothorax seem to show an elevated incidence.

In my opinion, you may better talk about "frequence". 

Reply: In principal both incidence and prevalence examine the frequency of illness or injury Incidence refers to the occurrence of new cases (frequency) of disease or injury in a population over a specified period. We used the same term used by the cited articles, which was “incidence”. In all the cited articles, incidence was checked, not prevalence. Incidence is more specific term than frequency. 

Second, when you argue about pneumothorax, you should speak about barotrauma. Barotrauma can be further composed of pneumothorax and pneumomediastinum.

It seems so uncommon to not have any pneumomediastinum in your population.

Up to date, what is known in COVID-19 is that pneumothorax is less frequent than pneumomediastinum compared to traditional ARDS. However, the effect of this phenomenon is largely unknown?

See this reference: McGuinness G, Zhan C, Rosenberg N, Azour L, Wickstrom M, Mason DM, Thomas KM, Moore WH, Increased Incidence of Barotrauma in Patients with COVID-19 on Invasive Mechanical Ventilation, Radiology, vol. 297, no. 2, pp. E252-E62, Nov 2020.doi:10.1148/radiol.2020202352.

The currently published scientific literature shows that pneumothorax occurs in 1 to 3% of hospitalized COVID-19 cases, with up to 6% in patients undergoing non-invasive ventilation (NIV) and mechanical ventilation (MV). Therefore, the frequency of pneumomediastinum and pneumothorax during COVID-19 is not well defined, as the available data are limited to case collections and single reports.

In McGuinness's analysis (see ref. above), which compared complications from barotrauma in patients with acute respiratory distress syndrome (ARDS) in VAM, the frequency of pneumothorax is 9%, of pneumomediastinum 10% in COVID- 19. In comparison, in non-COVID-19 patients, it is 12% for PNX and 3% for PMS.

Reply: Pneumomediastinum and subcutaneous emphysema occurs in the presence of alveolar micro injury that mostly is associated with pneumothorax or pneumothorax that is subclinical. We agree with your point. Therefore, we further examined our data and found that the incidence of pneumomediastinum was 20/334 (6%) and subcutaneous emphysema was 9/334 (3%). We added this finding in the result section. We focused on pneumothorax in our study as we feel it is more clinically important than pneumomediastinum or subcutaneous emphysema. 

Again, I agree with you that the true frequency of pneumothorax (both in COVID and ARDS) is not well documented. I am an intensivist with more than 20 years ICU experience and believe that 12% of pneumothorax is well an overestimation! Nonetheless, the discrepancies of reported frequencies for pneumomediastinum, and subcutaneous emphysema can be partially explained by the heterogeneity of the method used for diagnosis between the studies. For example, higher detection of pneumomediastinum or subcutaneous emphysema could result from increased use of chest CT in some studies. But usually in mechanically ventilated patients the pneumomediastinum and subcutaneous emphysema can evolve to pneumothorax.

2) INTRODUCTION:

This sentence needs a reference if true:

"It appears that the incidence of pneumothorax is increasing over time with subsequent waves of COVID-19 infection".

Reply. That you for your comment. We rephrase the sentence. As it is possible that in early phase of pandemic clinicians did not pay much attention to this complication. We added a reference to the above sentence: (Palumbo et al., 2021).

The referred study showed higher incidence of pneumothorax in the second wave of the pandemic compared to the first. We adjusted our manuscript to specifically mention this finding. 

Along the line, the authors use pneumothorax and barotrauma indifferently, which should be corrected.

Reply. Thank you for this observation, this was corrected. 

3) METHODS:

This section should be improved and explain inclusion and exclusion criteria and why the grade of pulmonary damage was not reported as lung ultrasound or CT scan scores.

Reply: We already explained in detail the inclusion and the exclusion criteria for our study sample.

Inclusion criteria included: “patients aged ≥ 18 with COVID-19 positive (with laboratory-confirmed SARS-CoV-2 infection), who were admitted to the intensive care unit (ICU) for respiratory distress at Detroit Receiving Hospital and Harper University Hospital in Detroit Medical Center in Detroit, Michigan from May 2020 to February 2021. 

While Exclusion criteria are: Patients with iatrogenic pneumothorax, or those requiring thoracentesis or chest tube drainage for pleural effusion; or receiving extracorporeal membrane oxygenation (ECMO) were excluded.

We also already provided a supplemental file for our Institutional guidelines for Tocilizumab therapy including a List of inclusion and exclusion criteria for tocilizumab therapy in our institution. 

Regarding the grade of pulmonary damage: unfortunately, only small number of patients had ultrasound or CT of the lung done. This is mainly due to limited resources during the pandemic. 

4) STATISTIC

In the statistical section, the power analysis of the first aim is missing.

Reply: Sorry I am not clear about the question. However, our initial intention was to evaluate the frequency of pneumothorax and evaluate the effect of gender. Based on power calculation (based on G power) we needed to have 148 subjects with evenly distributed gender to have a 95% CI. However, when we analyze our data, we saw no significant differences between gender. Therefore, we analyze based on the therapy and other characteristic such as ventilation. We did not mentioned the power analysis as does not contribute. 

5) RESULTS

"Subjects in the pneumothorax cohort were more likely to require prone positioning (75% vs 36%; p<0.001)" maybe this is the consequence, not the cause?

Reply: Thank you for comment. The sentence above did not establish a relationship between pneumothorax and prone and we did not mean to do that. We reporting that in the pneumothorax group, subjects required more prone position suggesting more hypoxia in pneumothorax group. We believe that pneumothorax is related to the severity of the disease not to proning. Reviewing the literature, Pneumothorax was never reported as a complication of proning (Prone Positioning of Patients with Acute Respiratory Distress Syndrome: A Systematic Review - ProQuest, n.d.). In our study, pneumothorax happened either after or before pronning (we added this to our manuscript). 

"(Table 1). Most patients developed pneumothorax while on invasive mechanical ventilation" however, in the two courts, there were no significant differences in ventilator parameters between the two groups, except the average RR. How do you make the diagnosis of barotrauma? Could barotrauma be present before intubation?

Reply: The diagnosis of pneumothorax was made based on chest X-ray results (This was added to the method section). At least once daily a CXR obtained during the ICU admission, in case of hypoxia or instability and after intubation we routinely perform CXR. Therefore, it is extremely unlikely that a patient developed pneumothorax before intubation and remained undetected by the ICU team. Regarding the time of pneumothorax related to intubation, we already mentioned this in our manuscript results section “Most patients developed pneumothorax while on invasive mechanical ventilation, except two patients, who developed pneumothorax while receiving HFNC or non-invasive mechanical ventilation”.

6) RESULTS:

"Additionally, the pneumothorax group was

more likely to receive anticoagulation therapy, 78% versus 58%, p=0.01" how do you modify therapy?

Reply: We added our institution guidance for COVID-19 specific therapies as well as anticoagulation in the supplemental file (S1 file). At that time, all patients with D-dimer >3 received full anticoagulation therapy (we also added this to the method section as well as to the result section). 

There were no significant differences in mortality. That is not easy to understand.

Reply: This is true! However, this is what our data showed. 

"Second, we investigated the risk factors of barotrauma using multivariate analysis and the Cox proportional regression hazard model and found that TCZ significantly increased the risk of pneumothorax in COVID-19 patients.

That is a secondary endpoint without power analysis, and an association does not mean that this is true.

Reply: We totally agree, association does not mean causation. We changed the sentence to “TCZ significantly associated with higher risk of pneumothorax in COVID-19 patients”. However, there is an statically significant association between TCZ and pneumothorax. Such association was not observed in case of steroid and other drugs. 

In this court, pneumothorax occurs relatively late (median 30 days) in the course

of the disease, when CRP and fever were already down-trending sound like, in the end, the lung has lost compliance and became more susceptible to barotrauma. We have observed in COVID-19 that barotrauma in COVID-19 is an early phenomenon linked to P-SILI without a clear explanation at the moment.

Reply: We think this is a great point. It is possible that the pathology of early versus late pneumothorax differs. After reviewing the literature, three studies found that barotrauma is a late complication of COVID-19 (Elsaaran et al., 2021) (Belletti et al., 2021) (Kahn et al., 2021). In these studies, the mean time to development of barotrauma was range between 10 to 18 days from intubation or admission. While other studies found that barotrauma developed early in the course of the disease (Kumar Swain et al., 2021) (Lemmers et al., 2020) (Özdemir et al., 2021). In these studies, the mean time to development of barotrauma was range between 3 to 5 days from intubation or admission. Because of late occurrence of pneumothorax, we think the wound healing process is likely affected rather than disease itself. However, our study can not determine the cause and effect. 

After reviewing our data and statistics, we found that the mean duration of mechanical ventilation prior to pneumothorax development was 10.3 days and the mean duration of ICU stay prior to pneumothorax was 15.7 days. Please keep in mind that most patients were admitted from the regular floor. This adds more days into the total hospital stay. This finding was added to the result section. 

Reviewer #2: Dear authors,

I enjoyed reviewing this interesting paper. The subject is interesting and the paper in well written. I have detailed some comments below.

Reply: We appreciate your comment and thank you for taking the time and effort necessary to review the manuscript.

MAJOR COMMENTS:

1. In the introduction, the authors state that "Pneumothorax occurs when alveoli become damaged and/or ruptured due to elevated peak and plateau pressures, and can manifest as spontaneous pneumothorax, subcutaneous emphysema and pneumomediastinum.". Nonetheless, this is not always true ( doi: 10.1007/s00134-004-2187-7). Moreover, during the COVID-19 pandemic there seemed to be a remarkable increase in pneumomediastinum/subcutaneous emphysema occurrence despite the use of the same protective mechanical ventilation protocol used in non- COVID ARDS patients (https://doi.org/10.1183%2F23120541.00385-2020).This sentence need therefore to be rephrased.

Reply: We agree with you. Our study also showed that PEEP, peak and plateau pressures were not significantly different between patients who developed pneumothorax and those who did not. We revised the writing.

2. Due to what is detailed in the above comment, the term "barotrauma" should be changed throughout the manuscript and abstract, using it only when elevated airway pressure is the cause of the presence of air outside the tracheobronchial tree. In its absence, such a condition should not be referred to as barotrauma, but simply described for what it is (pneumomediastinum, subcutaneous emphysema, pneumothorax)(https://doi.org/10.1183%2F23120541.00385-2020). In your study, tidal volume, PEEP, and peak and plateau pressures were not significantly different between patients who developed pneumothorax and those who did not.

Reply: We agree with you. This has been revised.

---

## [Editor Report · Decision Letter 1]

12 Jul 2022

Incidence and Risk Factors of COVID-19 Associated Pneumothorax

PONE-D-22-10994R1

Dear Dr. Samavati,

We’re pleased to inform you that your manuscript has been judged scientifically suitable for publication and will be formally accepted for publication once it meets all outstanding technical requirements.

Kind regards,

Andrea Cortegiani, M.D.

Academic Editor

PLOS ONE
---

## [Editor Report · Acceptance letter]

28 Jul 2022

PONE-D-22-10994R1 

Incidence and risk factors of COVID-19 associated pneumothorax 

Dear Dr. Samavati:

I'm pleased to inform you that your manuscript has been deemed suitable for publication in PLOS ONE. Congratulations! Your manuscript is now with our production department. 

Kind regards, 

on behalf of

Dr. Andrea Cortegiani 

Academic Editor

PLOS ONE